# Seeing Is Believing: Brain-Inspired Modular Training for Mechanistic Interpretability

**DOI:** 10.3390/e26010041

**Published:** 2023-12-30

**Authors:** Ziming Liu, Eric Gan, Max Tegmark

**Affiliations:** Institute for Artificial Intelligence and Fundamental Interactions, Massachusetts Institute of Technology, Cambridge, MA 02139, USA; ejgan@mit.edu (E.G.); tegmark@mit.edu (M.T.)

**Keywords:** brain-inspired artificial intelligence, mechanistic interpretability, modularity

## Abstract

We introduce Brain-Inspired Modular Training (BIMT), a method for making neural networks more modular and interpretable. Inspired by brains, BIMT embeds neurons in a geometric space and augments the loss function with a cost proportional to the length of each neuron connection. This is inspired by the idea of minimum connection cost in evolutionary biology, but we are the first the combine this idea with training neural networks with gradient descent for interpretability. We demonstrate that BIMT discovers useful modular neural networks for many simple tasks, revealing compositional structures in symbolic formulas, interpretable decision boundaries and features for classification, and mathematical structure in algorithmic datasets. Qualitatively, BIMT-trained networks have modules readily identifiable by the naked eye, but regularly trained networks seem much more complicated. Quantitatively, we use Newman’s method to compute the modularity of network graphs; BIMT achieves the highest modularity for all our test problems. A promising and ambitious future direction is to apply the proposed method to understand large models for vision, language, and science.

## 1. Introduction

Although deep neural networks have achieved great successes, mechanistically interpreting them remains quite challenging [1,2,3,4,5]. If a neural network can be decomposed into smaller modules [1], interpretability may become much easier.

In contrast to artificial neural networks, brains are remarkably modular [6,7,8]. We conjecture that this is because artificial neural networks (e.g., fully connected neural networks) have a symmetry that brains lack: both the loss function and the most popular regularizers are invariant under permutations of neurons in each layer. In contrast, the cost of connecting two biological neurons depends on how far apart they are because an axon needs to traverse this distance, thereby using energy and brain volume and causing time delay [7,8].

We argue that (current) artificial neural networks are missing a key ingredient, which is the very reason why human brains are modular. From Darwin’s evolution theory, modular brains have survival advantages over non-modular ones, since modular brains react faster by processing certain functions within local areas. By contrast, modular neural networks do not necessarily have “survival advantages” (e.g., lower losses) over non-modular ones in standard training.

To facilitate the discovery of more modular and interpretable neural networks, we introduce Brain-Inspired Modular Training (BIMT). Inspired by brains, we embed neurons in a geometric space where distances are defined and augment the loss function with a cost proportional to the length of each neuron connection times the absolute value of the connection weight. This obviously encourages *locality*, i.e., keeping neurons that need to communicate as close together as possible. Any Riemannian manifold can be used; we explore 2D and 3D Euclidean space for easy visualization (see Figure 1). Our work is inspired by the minimum connection cost idea explored in [9,10,11,12]. While their focus is on understanding biological neural networks under evolution, our goal is to enhance interpretability of artificial neural networks trained with gradient descent.

We demonstrate the power of BIMT on a broad range of tasks, finding that it can reveal interesting and sometimes unexpected structures. Qualitatively, BIMT-trained networks have modules readily visible to the naked eye, but standard trained networks seem to have much more complicated connections. Quantitatively, we use Newman’s method to compute modularity of the network connection graph; BIMT can achieve the highest modularity in all test cases. On symbolic formula datasets, BIMT is able to discover structures such as independence, compositionality, and feature-sharing, which are useful for scientific applications. For classification tasks, we find that BIMT may produce interpretable decision boundaries and features. For algorithmic tasks, we find BIMT to produce tree-like connectivity graphs, not only supporting the group representation argument in [13], but also revealing a (somewhat unexpected) mechanism where multiple modules vote. Although most of our experiments are conducted on fully connected networks for vector inputs, we also conduct experiments demonstrating that BIMT generalizes to other types of data (e.g., images) and architectures (e.g., transformers).

This paper is organized as follows: Section 2 introduces Brain-Inspired Modular Training (BIMT). Section 3 applies BIMT to various tasks, demonstrating its interpretability power. We describe related work in Section 4 and discuss our conclusions in Section 5.

## 2. Brain-Inspired Modular Training (BIMT)

Human brains are modular and sparse, which is arguably the reason why they are so efficient. To make neural networks more efficient, it is desirable to make them modular and sparse, just like our brains. Sparsity is a well-studied topic in neural networks and can be encouraged by including the $L_{1}/L_{2}$ penalty in training or by applying pruning to model weights [14,15]. As for modularity, most research explicitly introduce modules [16,17], but this requires prior knowledge about problem structures. Our motivation question is thus:
Q:What training techniques can induce modularity in otherwise non-modular networks?

In other words, our goal is to let modularity emerge from non-modular networks when possible. In this section, we propose a method called Brain-Inspired Modular Training (BIMT), which explicitly steers neural networks to become more modular and sparse during training. BIMT consists of three key ingredients (see Figure 1): (1) embedding the network in a geometric space; (2) training to encourage locality and sparsity; and (3) swapping neurons for better locality.

**Notation:** For simplicity, we describe how to perform BIMT with fully connected networks. Generalization to other architectures is possible. We distinguish between *weight layers* and *neuron layers*. Assuming a fully connected network to have *L* weight layers, whose *i*th weight layer (i=1,⋯,L) has weights $W_{i}\in R^{n_{i-1}\times n_{i}}$ and biases $b_{i}\in R^{n_{i}}$, where $n_{i-1}$ and $n_{i}$ are the number of neurons incoming to and outgoing from the *i*th weight layer, the *i*th (i=0,⋯,L) neuron layer has $n_{i}$ neurons. The input and output dimensions of the whole network are $n_{0}$ and $n_{L}$, respectively.

**Step 1: Embedding the network into a geometric space.** We now embed the whole network into a space where the *j*th neuron in the *i*th layer is the (i,j) neuron located at $r_{ij}$. If this is 2D Euclidean space, neurons in the same neuron layer share the same *y*-coordinate and are uniformly spaced in x∈[0,A](A>0). Different neuron layers are vertically separated by a distance $y_{*}>0$, so
(1)$$r_{ij}\equiv \left(x_{ij},y_{ij}\right)=\left(Aj/n_{i},iy_{*}\right).$$
The weight that connects the (i−1,j) neuron and the (i,k) neuron has value $w_{ijk}\equiv {\left(W_{i}\right)}_{jk}$, the bias at the (i,k) neuron is $b_{ik}\equiv {\left(b_{i}\right)}_{k}$, and its length is defined as
(2)$$d_{ijk}\equiv \left|r_{i-1,j}-r_{ik}\right|.$$
We will use $L_{1}$-norm, giving $d_{ijk}=\left|x_{i-1,j}-x_{ik}\right|+y_{*}$, but other vector norms can also be used. For example, $L_{2}$-norm gives $d_{ijk}={\left(|x_{i-1,j}-x_{ik}{|}^{2}+y_{*}^{2}\right)}^{1/2}$.

**Step 2: Imposing regularization that encourages locality.** We define the connection cost for weight and bias parameters of the whole network to be
(3)$$\ell^{w}=\sum_{i=1}^{L}\sum_{j=1}^{n_{i}}\sum_{k=1}^{n_{i+1}}d_{ijk}|w_{ijk}|,\;\ell^{b}=\sum_{i=1}^{L}\sum_{j=1}^{n_{i}}y_{*}\left|b_{ij}\right|.$$
When training for a particular task, in addition to the prediction loss $\ell^{\mathrm{pred}}$, we include $\ell^{w}$ and $\ell^{b}$ as regularizations:(4)$$\ell =\ell^{\mathrm{pred}}+\lambda \left(\ell^{w}+\ell^{b}\right),$$
where λ is the strength of the regularization. Without loss of generality, we can set $y_{*}=1$, leaving only the two hyper-parameters λ and *A*. Setting A=0 reduces to standard $L_{1}$ regularization, which solely encourages sparsity. A>0 further encourages locality, in addition to sparsity.

**Step 3: Swapping neurons for better locality.** We encourage locality (reduction of $\ell^{w}$) not only by updating weights via gradient descent, but also by swapping two neurons in the same neuron layer (i.e., swapping corresponding incoming/outgoing weights), when this reduces $\ell^{w}$. Gradient descent (continuous search) can get stuck at bad local minima where non-local connections are still present (see Figure 2c), while swapping (discrete search) can avoid this (see Figure 2e). Such swapping leaves the function implemented by the whole network (hence, $\ell^{\mathrm{pred}}$) unchanged but improves locality (see Figure 1, right). However, trying every possible permutation is prohibitively expensive. We assign each neuron (i,j) a score $s_{ij}$ to indicate its importance:(5)$$s_{ij}=\sum_{p=1}^{n_{i-1}}|w_{ipj}|+\sum_{q=1}^{n_{i+1}}\left|w_{i+1,jq}\right|,$$
which is the sum of (absolute values) of incoming and outgoing weights. We sort neurons in the same layer based on their scores and define neurons with the top *k*-scores as “important” neurons. For each important neuron, we swap it with the neuron in the same layer, causing the greatest decrease in $l^{w}$ if it helps. Since swaps are somewhat expensive, requiring O(nkL) computations, we implement swaps only every S≫1 training steps. We allow swaps also of input and output neurons, if not stated otherwise.

**BIMT = $L_{1}$ + Local + Swap.** To summarize, BIMT means local $L_{1}$ regularization with swaps. Both “local” and “swap” are novel contributions of this paper, while $L_{1}$ regularization is quite standard. If one wants to ablate “local” or “swap”, one can set A=0 to remove “local” or set S→∞ to remove “swap”. Our experience is that the joint use of “local” and “swap” usually gives the most interpretable networks. As a simple case, we compare BIMT to baselines on a regression problem, shown in Figure 2. On top of $L_{1}$, although using “local” or “swap” alone gives reasonably interpretable networks, the joint use of both produces the most interpretable network (at least visually). Although using $L_{1}$ alone leads to a reasonably sparse network, the network is neither modular nor optimally sparse (see Appendix A for pruning results).

**Connectivity Graphs.** As in Figure 2 and throughout the paper, we use connectivity graphs to visualize neural network structures. For visualization purposes, we normalize weights by the max absolute value in the same layer (so the normalized values lie in range [−1,1]). A weight is displayed as a line connecting two neurons, with its thickness proportional to its normalized value, and its color set to blue (red) if the value is positive (negative). Note that we draw all weights and do not explicitly ignore small weights. The reason connectivity graphs appear sparse is because naked eyes cannot identify very thin lines.

## 3. Experiments

In this section, we apply BIMT to a wide range of applications. In all cases, BIMT can result in modular and sparse networks, which immediately provide interpretability on the microscopic level and the macroscopic level. At the microscopic level, we can understand which neurons are useful, what each useful neuron is doing, and where/how information of interest is located/computed. At the macroscopic level, we can understand relations between different modules (e.g., in succession or in parallel) and how they cooperate to make the final prediction. From Section 3.1, Section 3.2 and Section 3.3, we train fully connected neural networks with BIMT for regression, classification, and algorithmic tasks. In Section 3.4, we show that BIMT can generalize to transformers and demonstrate it through in-context learning. In Section 3.5, we demonstrate that BIMT can easily go beyond vector-type data to tensor-type data (e.g., images). In general, BIMT achieves interpretability with either no drop or a modest drop in performance, summarized in Table 1. We also report the modularity metric in Table 2, computed using Newman’s method provided by the NetworkX package, showing that BIMT gives the most modular networks in all examples. Our baseline training methods are training with no penalty or with the $L_{1}$ [18] penalty with the Adam optimizer [19]. Results are averaged over five random seeds. All experiments are runnable on a CPU (M1), usually within minutes (at most two hours).

### 3.1. Symbolic Formulas

Symbolic formulas are prevalent in scientific domains. In recent years, as increasingly more data are collected from experiments, it is desirable to distill symbolic formulas from experimental data, a task called symbolic regression [20]. However, symbolic regression usually faces an all-or-nothing situation, i.e., either succeeds gloriously or fails miserably. Consequently, a tool supplementary to symbolic regression is called for, which can robustly reveal the high-level structure of formulas. We show below that BIMT can discover such structures in formulas.

We consider the task of predicting $y=(y_{1},\cdots ,y_{d_{o}})$ from $x=(x_{1},\cdots ,x_{d_{i}})$, where $y_{i}=f_{i}\left(x\right)$ are symbolic functions. We randomly sample each $x_{i}$ from U[−1,1] and compute $y_{i}=f_{i}\left(x\right)$ to generate the dataset. We use fully connected networks with SiLU activations (architectures shown in Figure 3) and training networks using the MES loss with the Adam optimizer with learning rate $10^{-3}$ for 20,000 steps, while choosing A=2, $y_{*}=0.1$, k=6, and S=200. We schedule λ as such: $\left(10^{-3},10^{-2},10^{-3}\right)$ for (5000, 10,000, 5000) steps.

We apply BIMT to several formulas, each of which has certain modular properties, as shown in Figure 3. (a) **Independence**. $y_{1}=x_{2}^{2}+\sin\left(\pi x_{4}\right)$ is independent of $x_{1}$ and $x_{3}$, while $y_{2}={\left(x_{1}+x_{3}\right)}^{3}$ is independent of $x_{2}$ and $x_{4}$. As desired, BIMT results in a network split into two parallel modules independent of each other, one only involving $\left(x_{1},x_{3}\right)$, the other only involving $\left(x_{2},x_{4}\right)$. (b) **Feature Sharing**. For targets $\left(y_{1},y_{2},y_{3}\right)=\left(x_{1}^{2},x_{1}^{2}+x_{2}^{2},x_{1}^{2}+x_{2}^{2}+x_{3}^{2}\right)$, learning shared features $\left(x_{1}^{2},x_{2}^{2},x_{3}^{2}\right)$ is beneficial for predicting all targets. Indeed, in the neuron layer A2, the only three active neurons correspond to these shared features (see Appendix B). (c) **Compositionality**. Computing $y=\sqrt{{\left(x_{1}-x_{2}\right)}^{2}+{\left(x_{3}-x_{4}\right)}^{2}}\equiv \sqrt{I}$ requires computing *I* first, which is an important intermediate quantity. We find that the only active neuron in layer A3 has activations highly correlated with *I*. Although one might worry that these extremely sparse networks could severely under-fit, we show that fitting is reasonably good in Appendix B.

### 3.2. Two Moon Classification

Interpretable decision boundaries help to make classification trustworthy. Moreover, decision boundaries with fewer pieces are more likely to generalize better. Therefore, it is desirable that neural networks used for classifications are sparse and interpretable.

We apply BIMT to the toy Two Moon dataset [21]. The architecture is shown in Figure 4 (the final softmax layer is not shown), with the same training details used in Section 3.1, with the only difference being the use of cross-entropy loss. We choose λ=0.01 and A=2. The evolution of the neural network is shown in Figure 4. Starting from a (randomly initialized) dense network, the network becomes increasingly sparse and modular, ending up as a network with only six useful hidden neurons. We can roughly split the training process into three phases: (i) in the first phase (steps 0 to 1000), the neural network mainly aims to fit the data while slightly sparsifying the network; (ii) in the second phase (steps 1000 to 3000), the neural network sparsifies the network in a symmetric way (both outputs of Classes 1 and 2 have neurons connecting to them); (iii) in the third phase (steps 3000 to end), the network prunes itself to become asymmetric, with useful neurons only connecting to Class 1 output. In Appendix C, we interpret what each weight is doing by editing them (zeroing) and see how this affects decision boundaries.

### 3.3. Algorithmic Datasets

Algorithmic datasets are ideal for benchmarking mechanistic interpretability methods because they are mathematically well understood. Consider a binary operation a∘b=c (a,b,c are discrete and treated as tokens), in which a neural network is tasked with predicting *c* from embeddings of *a* and *b*. For modular addition, Ref. [22] discovers that ring-like representations emerge in training. Ref. [23] reverse-engineered these networks, finding that the network internally implements trigonometric identities. For more general group operations, Ref. [13] suggests that representation theory is key for neural networks to generalize. However, in these papers, it is usually not obvious which neurons are useful or what the overall modular structure of the network is. Since BIMT explicitly optimizes modularity, it is able to produce networks that self-reveal their structure.

**Modular addition.** The task is to predict *c* from (a,b), where a+b=c(mod59). Each token *a* is embedded as a d=32-dimensional vector $E_{a}$, initialized as a random normal vector at initialization and trainable later. The concatenation of $E_{a}$ and $E_{b}$ is fed to a two-hidden-layer MLP, shown in Figure 5. We split train/test 80/20%. We train the network with BIMT with cross-entropy loss using the Adam optimizer (lr $=10^{-3}$) for 20,000 steps. We choose A=2, $y_{*}=0.5$, k=30, and S=200. We schedule λ as such: (0.1,1,0.1) for (5000, 10,000, 5000) steps.

After training, the network looks like a tree with three roots (A, B, C), shown in Figure 5. We visualize embeddings corresponding to these roots (modules), finding that the token embeddings form circles in 2D (A, B) and a bow tie in 3D (C). In contrast to Refs. [22,23], where post-processing (e.g., principal component analysis) is needed to obtain ring-like representations, the ring structures here automatically align to privileged bases, which is probably because embeddings are also regularized with $L_{1}$. To evaluate how these parallel modules are important for making predictions, we compute accuracy after knocking out some of them. The result is quite surprising: knocking out one of the modules can severely degrade the performance (from 100% to 15.25%, 29.33%, or 33.67% for knocking out A, B, or C). This means that modules are cooperating together to make predictions correct, similar to majority voting for error correction. To verify the universality of this argument, we include more tree graphs for perturbed initializations and different random seeds in Appendix D.

**Permutation group.** The task is to predict *c* from (a,b), where a,b,c are elements in the 24-element group (the permutation of 4 objects) $S_{4}$, and ab=c. Our training is the same as for modular addition. Figure 6 shows that, after training with BIMT, the network is quite modular. Notice that there are only nine active components in the embedding layer, exactly agreeing with the representation theory argument of Ref. [13] ($S_{4}$ has a 3×3 matrix representation). In Figure 6 (right), we show how each embedding neuron is activated by each group element, revealing that BIMT has discovered a crucial group-theoretical structure! Note that we have normalized these embeddings when plotting: denote the value of the *i*th neuron and the *j*th token as $e_{ij}$. The normalized embedding is defined as ${\tilde{e}}_{ij}=e_{ij}/\left(\max_{j}\left|e_{ij}\right|\right)$. In particular, neuron 22 is the sign neuron (1/−1 for even/odd permutations), and other active neurons correspond to subgroups or cosets (see further analysis in Appendix E).

### 3.4. Extension to Transformers: In-Context Linear Regression

So far, we have demonstrated the effectiveness of BIMT for MLPs. We can generalize BIMT to transformers [24]: we simply apply BIMT to linear layers in transformers (see details in Appendix F). Following the setup of Ref. [25], we now study in-context linear regression. Linear regression aims to predict *y* from $x\in R^{d}$, assuming that we know training data $\left(x_{i},y_{i}\right)\;\left(i=1,\cdots ,n\right)$, where $y_{i}=w\cdot x_{i}$. In-context linear regression aims to predict *y* from the sequence $\left(x_{1},y_{1},\cdots ,x_{n},y_{n},x\right)$, which is called in-context learning because the unknown weight vector w needs to be learned in context, i.e., when the transformer runs in test time rather than when it is trained. To make things maximally simple, we choose d=1 (the weight vector degrades to a scalar) and n=1.

The architecture is displayed in Figure 7, where, for clarity, we only show the last block, ignoring its attention dependence on previous blocks. The embedding size is 32, the number of transformer layers is 2 (each layer containing an attention layer and an MLP), and the number of heads is 1. We draw w∈U[1,3] (Instead, we can investigate w∼U[−1,1], which has a singularity issue (please see Appendix F for details).) and x∈U[−1,1] to create datasets. With MSE loss, we train with the Adam optimizer (lr: $1\times 10^{-3}$) for $4\times 10^{4}$ steps (λ=0.001,0.01,0.1,0.3 each for $10^{4}$ steps). We choose A=2, $y_{*}=0.5$, k=30, and S=200.

It is shown in Ref. [25] that *w* is linearly encoded in neural network latent representations, but it is not easy to track where this information is located. From Figure 7 (left), it is immediately clear which neurons are useful (active). In Figure 7 (top right), we show that the prediction is quite good, even though the network has become extremely sparse. We examine active neurons in the Res2 layer, finding that several neurons are correlated with the weight scalar, although no one neuron alone can determine the weight scalar perfectly. In Figure 7 (right middle and bottom), we show that pairs of neurons (8 and 9, 11 and 19) implicitly encode information about the weight scalar in nonlinear ways.

### 3.5. Extension to Tensor Data: Image Classification

So far, we have always embedded neural networks into 2D Euclidean space, but BIMT can be used in any geometric space. We now consider a minimal extension: embedding neural networks into a 3D Euclidean space. For 2D image data, to maintain their local structure, it is better to leave them as 2D rather than flatten them to 1D. As a result, an MLP for 2D image data should be embedded in 3D, as shown in Figure 8. The only modification for BIMT is that, when computing distances, we use 3D rather than 2D vector norms.

We train with MSE loss and use the Adam optimizer (lr = $1\times 10^{-3}$) for $4\times 10^{4}$ steps (λ=0.001,0.01,0.1,0.3 each for $10^{4}$ steps). We choose A=2, $y_{*}=0.5$, k=30, and S=200. We disable swaps of input pixels. We show the evolution of the network in Figure 8. Starting from a dense network, the network becomes more modular and sparser over time. Notably, the receptive field shrinks for the input layer, since BIMT learns to prune away peripheral pixels that always equal zero. Another interesting observation is that most of the weights in the middle layer are negative (colored red), while most of the weights in the last layer are positive (colored blue). This suggests that the middle layer is not adopting the strategy of pattern matching, but rather *pattern mismatching*. Pattern matching/mismatching means that, if an image has/does not have these patterns, it is more likely to be, for example, an 8. We visualize features in Appendix G, where we also include the results for MLPs with different depths. Moreover, in the output layer, Classes 1 and 7 are automatically swapped to become neighbors, probably due to their similarity. In future work, we would like to compare our method with convolutional neural networks (CNNs). It might be best to combine CNNs with BIMT, since CNNs guarantee the locality of inputs, while BIMT encourages locality of model internals.

## 4. Related Work

**Mechanistic Interpretability** (MI) is an emerging field that aims to mechanically understand how neural networks work. Various modules/circuits are identified from neural networks via reverse engineering, including image circuits [1], induction heads [2], computational quanta [3], transformer circuits [4], factual associations [26], and heads in the wild [5], although superposition [27] makes interpretability more complicated. A generalization puzzle called grokking [28] has also been understood by reverse-engineering neural networks [13,22,23,29].

**Modularity** in neural networks can help generalization in transfer learning [16] and can enhance interpretability [1]. Non-modular neural networks trained in standard ways are shown to present an imperfect extent of modularity [30,31,32]. Modular networks explicitly use trainable modules in constructing neural networks [17,33], but this inductive bias may require prior knowledge about the tasks. The multi-head attention layer in transformers lies in the category of explicitly introducing modularity. By contrast, this work does not explicitly introduce modules, but rather lets modules emerge from non-modular networks with the help of BIMT.

**Pruning** can lead to sparse and efficient neural networks [14,15,34,35], usually achieved by $L_{1}$ or $L_{2}$ regularization and thresholding small weights to zero. BIMT borrows the $L_{1}$ regularization technique for sparsity but improves modularity by making the $L_{1}$ regularization distance-dependent.

**Analogy between neuroscience and neural networks** has existed for a long time in the literature [36,37]. Although biological and artificial neural networks may not have the same low-level learning mechanisms [38], we can still borrow high-level ideas and concepts from neuroscience to design more interpretable artificial neural networks, which is the goal of this work. The minimal connection cost idea has been explored in [9,10,11,12], where an evolutionary algorithm is applied to evolve tiny networks. By contrast, our method is more aligned with modern machine learning, i.e., gradient-based optimization and broader applications.

**Neuroscience-inspired learning.** Since the literature of neuroscience is vast [39], we will not review it here, but we do want to highlight some progress in neuroscience that can potentially inspire and improve current machine learning systems. The study of neural codes [40], neural information processing and transmission [41,42,43], and neural spikes [44] may provide tools and language towards the study and design of artificial neural networks.

## 5. Conclusions and Discussion

We have proposed Brain-Inspired Modular Training (BIMT), which explicitly encourages neural networks to be modular and sparse. BIMT is a principled idea that could be generalized to many types of data and network architectures. Tested on several relatively small-scale tasks, we show its ability to provide interpretable insights for these problems. In future studies, we would like to see if this training strategy remains valid for larger-scale tasks, e.g., large language models (LLMs). In particular, can we fine-tune LLMs with BIMT to make them more interpretable? Moreover, BIMT achieves interpretability at the price of slight performance degradation. In fact, it is known that modularity might improve performance in continual learning when the tasks have modular and compositional structures [12]. To understand the benefits and limitations of BIMT, further studies will require careful disentanglement of network sparsity and modularity and rigorous definition of task modularity. We would like to improve BIMT such that interpretability and performance are achieved at the same time.

**Broader Impacts.** We believe that building interpretable neural networks will make AI more controllable, more reliable, and safer. However, like other AI interpretability research, the controllability brought by interpretability should be regulated, making sure the technology is not misused.

**Limitations.** (1) This work deals with small-scale toy problems, where neural networks can be easily visualized. It is still unclear whether this method remains effective for larger-scale problems. (2) BIMT requires users to define the embedding geometric space, which may require some prior knowledge about the task. Arguably, this might be a feature not a bug, especially for neuromorphic computing. (3) The swapping step may incur additional overhead. Additionally, swapping is currently implemented layer by layer, so global topological problems cannot be resolved via swapping. (4) Visualizing the connectivity graph of neural networks is only efficient and useful for small networks. (5) There is generally a trade-off between accuracy and simplicity/interpretability, which is also true for BIMT. (6) The connectivity graphs can be quite sensitive to random seeds, although they share common global structures, which might be enough for interpretation. Please see Appendix D for more details.

**Future Directions.** Two major concerns of this work are scalability and sensitivity. We would like to mention our plans for how to address these two issues in future work. To improve scalability, we may need to explicitly introduce some notion of hierarchy (beyond modularity), because hierarchy can potentially decompose complicated modules into simpler sub-modules, hence making scaling easier. To reduce sensitivity, some theoretical research is first needed to characterize equivalent classes of neural networks so that we can better understand whether the changes caused by perturbation are superficial (in the same equivalent class) or fundamental (in different equivalent classes).

## Figures and Tables

**Figure 1 entropy-26-00041-f001:**
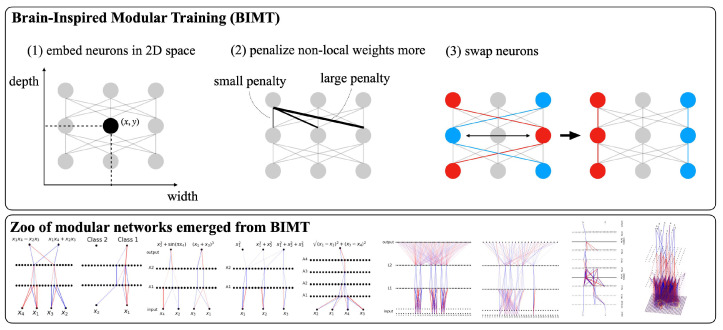
(**Top**): Brain-Inspired Modular Training (BIMT) contains three ingredients: (1) embedding neurons into a geometric space (e.g., 2D Euclidean space); (2) training with regularization, which penalizes non-local weights more; (3) swapping neurons during training to further enhance locality. (**Bottom**): Zoo of modular networks obtained via BIMT (see experiments for details).

**Figure 2 entropy-26-00041-f002:**
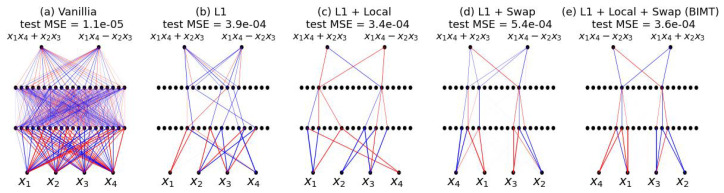
The connectivity graphs of neural networks when trained with different techniques for a regression problem (blue/red denote positive/negative weights). Our proposed BIMT = $L_{1}$ regularization (not novel) + local regularization (novel) + swap (novel). BIMT finds the simplest circuit (**e**) which clearly contains two parallel modules, with a moderate sacrifice in test loss compared to vanilla (**a**), but with lower loss than for mere $L_{1}$ regularization (**b**). Note that swapping aims to reduce the local connection cost, so all of (**c**–**e** ) encourage locality.

**Figure 3 entropy-26-00041-f003:**
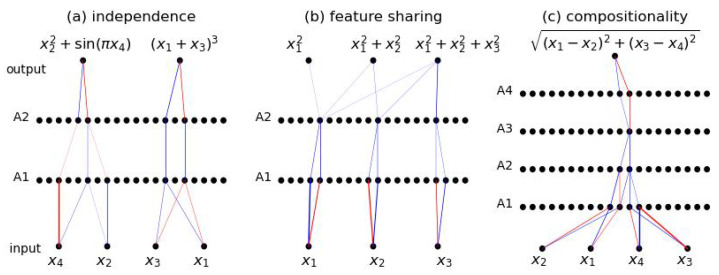
The connectivity graphs of neural networks trained with BIMT to regress symbolic formulas (blue/red lines stand for positive/negative weights). For symbolic formulas with modular properties, e.g., independence, shared features, or compositionality, the connectivity graphs display modular structures revealing these properties.

**Figure 4 entropy-26-00041-f004:**
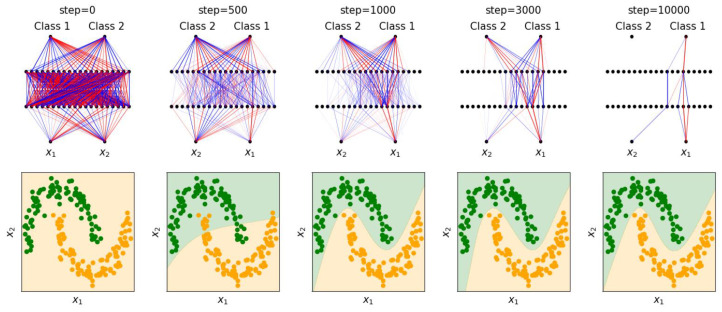
(**Top**): Evolution of network structures trained with BIMT on the Two Moon dataset. Blue and red lines stand for postive and negative weights, respectively. (**Bottom**): Evolution of decision boundaries.

**Figure 5 entropy-26-00041-f005:**
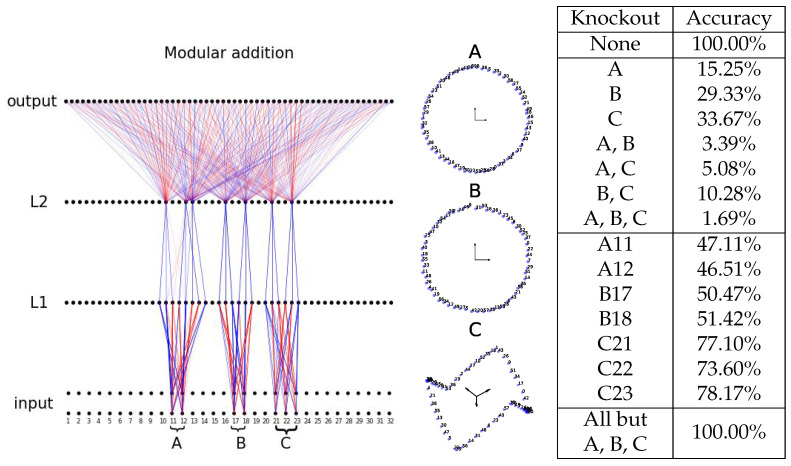
MLP trained with BIMT for modular addition. (**Left**): the final connectivity graph is tree-like, demonstrating three parallel modules (voters); middle: the representations of each module in the input layer. Blue and red lines stand for postive and negative weights, respectively. (**Right**): ablation results, which imply a voting mechanism. The input layer contains embeddings of two tokens, which overlap each other but are drawn to be vertically separated.

**Figure 6 entropy-26-00041-f006:**
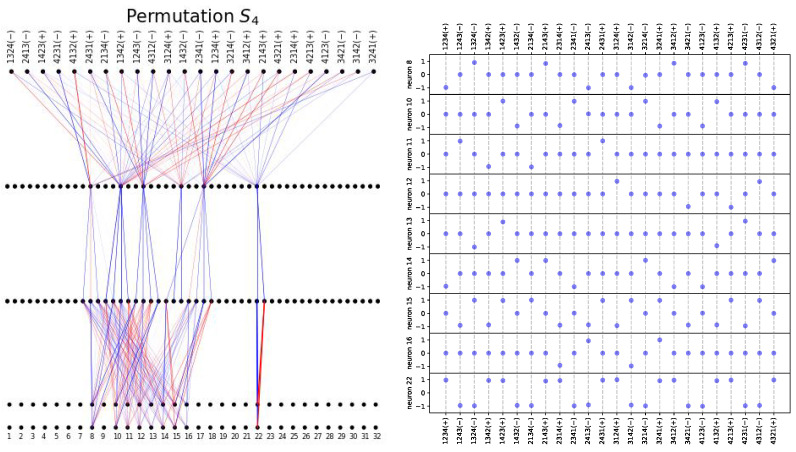
Apply BIMT to MLP on the Permutation $S_{4}$ dataset. (**Left**): the final connectivity graph, with only nine active embedding neurons. The input layer contains embeddings of two tokens, which overlap each other but are drawn to be vertically separated. Blue and red lines stand for postive and negative weights, respectively. (**Right**): the nine active neurons correspond to group representations of $S_{4}$, whose values are normalized into the range [−1,1]. In particular, neuron 22 is the sign neuron (1/−1 for even/odd permutations).

**Figure 7 entropy-26-00041-f007:**
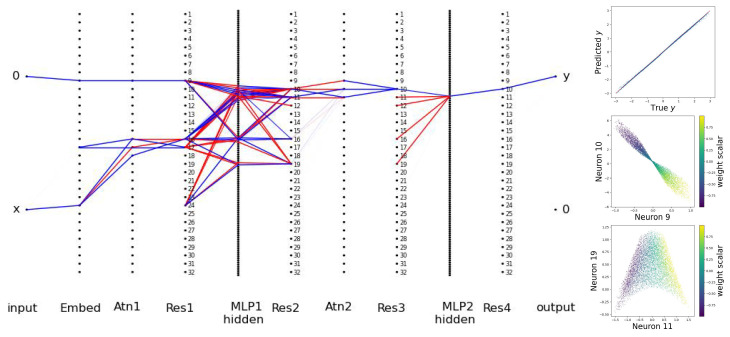
Application of BIMT to transformers during in-context learning linear regression. Left: the connectivity graph of the transformer after training. Only the last block is shown, which takes in [0,x] to predict [y,0]. Blue and red lines stand for postive and negative weights, respectively. Right top: predicted vs true *y*. Right middle and bottom: neurons in the Res2 layer contain the information about the weight scalar, encoded non-linearly.

**Figure 8 entropy-26-00041-f008:**
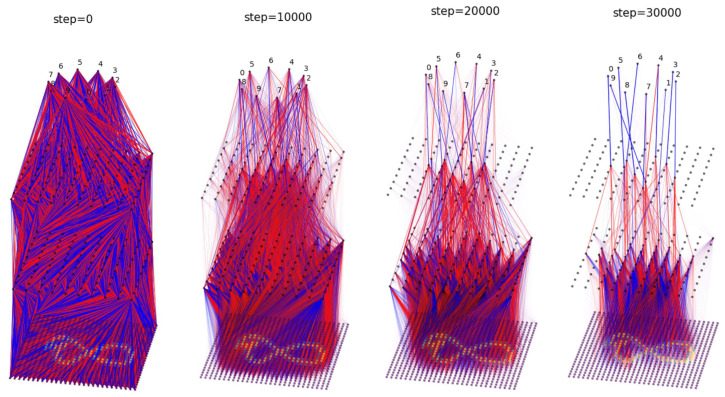
Application of BIMT to 3D MLP on MNIST. From left to right: connectivity graph evolution. Blue and red lines stand for postive and negative weights, respectively.

**Table 1 entropy-26-00041-t001:** BIMT achieves interpretability with no or a modest performance drop.

Dataset	Symbolic (a)	Symbolic (b)	Symbolic (c)	Two Moon	Modular Addition	Permutation	In-Context	MNIST
metric	loss	loss	loss	accuracy	accuracy	accuracy	loss	accuracy
Vanilla	$\left(5.2\pm 1.0\right)\times 10^{-3}$	$\left(1.1\pm 0.4\right)\times 10^{-5}$	$\left(1.2\pm 0.5\right)\times 10^{-4}$	(100.0±0.0)%	(100.0±0.0)%	(100.0±0.0)%	$\left(7.8\pm 1.8\right)\times 10^{-5}$	(98.5±0.2)%
$L_{1}$ penalty	$\left(7.9\pm 0.8\right)\times 10^{-3}$	$\left(1.2\pm 0.3\right)\times 10^{-5}$	$\left(1.8\pm 0.4\right)\times 10^{-4}$	(100.0±0.0)%	(100.0±0.0)%	(100.0±0.0)%	$\left(7.2\pm 1.0\right)\times 10^{-5}$	(98.4±0.3)%
BIMT (ours)	$\left(7.4\pm 1.0\right)\times 10^{-3}$	$\left(8.0\pm 1.5\right)\times 10^{-5}$	$\left(1.3\pm 0.3\right)\times 10^{-3}$	(100.0±0.0)%	(100.0±0.0)%	(100.0±0.0)%	$\left(1.8\pm 0.4\right)\times 10^{-4}$	(98.2±0.3)%

**Table 2 entropy-26-00041-t002:** BIMT achieves the highest modularity for all tasks.

Dataset	Symbolic (a)	Symbolic (b)	Symbolic (c)	Two Moon	Modular Addition	Permutation	In-Context	MNIST
Vanilla	0.207±0.025	0.151±0.015	0.145±0.023	0.291±0.032	0.194±0.028	0.129±0.012	0.085±0.014	0.073±0.005
$L_{1}$ penalty	0.512±0.042	0.456±0.035	0.328±0.027	0.289±0.016	0.435±0.028	0.371±0.031	0.083±0.010	0.099±0.008
BIMT (ours)	0.581±0.042	0.543±0.063	0.392±0.021	0.338±0.039	0.535±0.027	0.634±0.043	0.161±0.013	0.238±0.021

## Data Availability

The data presented in this study can be reproduced with code openly available at https://github.com/KindXiaoming/BIMT accessed on 1 November 2023.

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
