# Peer review of "Seeing Is Believing: Brain-Inspired Modular Training for Mechanistic Interpretability"

_entropy, 2023, doi:10.3390/e26010041_

Round 1
Reviewer 1 Report
Comments and Suggestions for Authors
This study proposed that including the connection cost in the loss function of feed-forward neural networks could simplify the network structure after training and improve mechanistic interpretability. The Brain-Inspired Modular Training (BIMT) was tested in several small-scale simple tasks. In all tasks, sparse straightforward networks were obtained after BIMT, which provided us relatively interpretable mapping processes. The concept was very clear, and many results supported its claims. However, there is concern that the application of BIMT would degrade the performance. In addition, the definition of modularity was unclear, which may confuse readers. Major comments are as follows.
(1) Define modularity. In graph theory, the modularity is a measure the strength of division of a network into modules. Networks with high modularity have dense connections between the nodes within modules but sparse connections between nodes in different modules. Does this definition apply to this paper?
(2) Quantify modularity. If the above definition is correct, it is questionable whether some of the resulting network structures have modules. How many modules did they have and what is the degree of modularity? For example, use Newman’s method to estimate the modularity of the resulting networks.
(3) Refer the related studies in Introduction. References [37-40], especially [40] are very relevant to this study. Please review them and clarify how they differ from this study in Introduction.
(4) Is table 1 the results of one trial? Please show the average results of multiple trials with different random seeds.
(5) The usefulness of BIMT is questionable as its use reduces performance. Are there cases where performance is improved by the networks having modules? Reference [40] demonstrated the modularity might improve the performance in continual learning. At least, discuss such cases.
(6) Indicate the parameter settings. BIMT has two important hyper parameters: lambda and A. Please specify the values of the parameters in each case to reproduce the experiments.
Author Response
We would like to thank the reviewer for their time and constructive suggestions! We agree with all the suggestions and have revised the paper accordingly. Below is our response to each question:
Q1 & Q2: Define and quantify modularity.
A: Yes! The modularity metric in graph theory applies to our case. Indeed, we hope: (1) there are dense connections within modules such that these module have sufficient expressive power; (2) there are sparse connections between different modules such that the relation between modules is interpretable. We're now using Newman's method to compute modularity, by calling the modularity function in the package NetworkX. Results are reported in Table 2 of the updated manuscript. Our method BIMT achieves the highest modularity for all tasks, compared with Vanilla training (no penalty) and L1 penalty.
Q3: Mention related works in introduction.
A: Thanks for the suggestion! We now include this in the introduction to clarify their differences: "Our work is inspired by the minimum connection cost idea explored in [37-40]. While their focus is understanding biological neural networks under evolution, our goal is to enhance interpretability of artificial neural networks trained with gradient descent."
Q4: Try multiple trials.
A: Thanks for the suggestion! We're now running 5 random seeds and report their means and standard deviations in Table 1 & Table 2. It turns out that there are indeed fluctuations between different runs, but the advantages of BIMT over baseline methods still stand despite such randomness.
Q5: BIMT slightly harms performance, but there are cases where modularity might improve performance in the continual learning scenario.
A: Agreed! We have added this discussion in conclusion: "Moreover, BIMT achieves interpretability at the price of slight performance degradation. In fact, it is known that modularity might improve performance in continual learning when the tasks have modular and compositional structures [40]. To understand the benefits and limitations of BIMT, further studies will require careful disentanglement of network sparsity and modularity, and rigorous definition of task modularity. We would like to improve BIMT such that interpretability and performance are achieved at the same time.". How to measure the benefits and limitations of modularity can be quite case-dependent and subtle: (1) whether the task(s) have modular structures matter; (2) the choice of the performance metric matters (e.g., hard or soft metrics may behave differently); (3) the correlation between sparsity and modularity matters. BIMT encourages both sparsity and modularity, so it is unclear whether performance degradation comes from sparsity or solely from modularity. We would like to investigate these issues in future works.
Q6: BIMT hyperparameters lambda and A.
A: Thanks! We have now added or highlighted lambda and A for each example.
Reviewer 2 Report
Comments and Suggestions for Authors
The Authors proposed brain-inspired modular training (BIMT), which explicitly encourages neural networks to be modular and sparse. That paper is interesting and quite well-written. However, I have the following comments and concerns:
1. The paper abstract is written very generally. The paper abstract must be reformulated to highlight the added value of the paper (including what measurable effects the Authors achieved - the quantitative value of the proposed approach should be included).
2. The introduction to the topic is not sufficient. It is very general, this issue requires expansion.
3. The lack of reference to other learning methods.
4. In my opinion, a brief paragraph about the fundamental problems in Neurosciences, specifically the problems concerning the form of neuronal codes (neural communication) should be also added. This would also give some insight into why precise spike detection is so important. Here are a few fundamental and recent papers that could be useful for the preparation of such paragraph:
-van Hemmen, J. L.; Sejnowski, T., 23 Problems in Systems Neurosciences, Oxford University Press, Oxford, 2006.
-Rieke, F.; Warland, D. D.; de Ruyter van Steveninck, R. R.; Bialek, W. Spikes: Exploring the Neural Code, MIT Press, 1997.
-Pregowska, A. Signal Fluctuations and the Information Transmission Rates in Binary Communication Channels. Entropy 2021, 23, 92.
-Mainen, Z. F.; Sejnowski, T. J., Reliability of spike timing in neocortical neurons. Science 1995, 268 (5216), 1503-1506.
-Salinas, E.; Sejnowski, T. J., Correlated neuronal activity and the flow of neural information. Nature Reviews Neuroscience 2001, 2, 539-550.
-Knoblauch, A.; Palm, G., What is signal and what is noise in the brain? Biosystems 2005, 79 (1–3), 83-90.
5. There is practically no discussion in the paper, and no reference is made to other results (other articles). This issue needs to be definitively improved. Please, look at the instructions for Authors.
6. What are the practical implications of the proposed approach?
7. All Figures are of very poor quality.
Thus, this paper is very interesting and the results obtained are of some importance. However, due to the above comments, I would recommend the article for publication, provided that the above concerns will be addressed. I recommend a Major Revision.
The Authors proposed brain-inspired modular training (BIMT), which explicitly encourages neural networks to be modular and sparse. That paper is interesting and quite well-written. However, I have the following comments and concerns:
1. The paper abstract is written very generally. The paper abstract must be reformulated to highlight the added value of the paper (including what measurable effects the Authors achieved - the quantitative value of the proposed approach should be included).
2. The introduction to the topic is not sufficient. It is very general, this issue requires expansion.
3. The lack of reference to other learning methods.
4. In my opinion, a brief paragraph about the fundamental problems in Neurosciences, specifically the problems concerning the form of neuronal codes (neural communication) should be also added. This would also give some insight into why precise spike detection is so important. Here are a few fundamental and recent papers that could be useful for the preparation of such paragraph:
-van Hemmen, J. L.; Sejnowski, T., 23 Problems in Systems Neurosciences, Oxford University Press, Oxford, 2006.
-Rieke, F.; Warland, D. D.; de Ruyter van Steveninck, R. R.; Bialek, W. Spikes: Exploring the Neural Code, MIT Press, 1997.
-Pregowska, A. Signal Fluctuations and the Information Transmission Rates in Binary Communication Channels. Entropy 2021, 23, 92.
-Mainen, Z. F.; Sejnowski, T. J., Reliability of spike timing in neocortical neurons. Science 1995, 268 (5216), 1503-1506.
-Salinas, E.; Sejnowski, T. J., Correlated neuronal activity and the flow of neural information. Nature Reviews Neuroscience 2001, 2, 539-550.
-Knoblauch, A.; Palm, G., What is signal and what is noise in the brain? Biosystems 2005, 79 (1–3), 83-90.
5. There is practically no discussion in the paper, and no reference is made to other results (other articles). This issue needs to be definitively improved. Please, look at the instructions for Authors.
6. What are the practical implications of the proposed approach?
7. All Figures are of very poor quality.
Thus, this paper is very interesting and the results obtained are of some importance. However, due to the above comments, I would recommend the article for publication, provided that the above concerns will be addressed. I recommend a Major Revision.
Author Response
We would like to thank the reviewer for their time and constructive suggestions! We agree with all the suggestions and have revised the paper accordingly. Below is our response to each question:
Q1: Abstract is written too generally.
A: Agreed! We have revised the abstract to make it more specific and concrete. Regarding added value: "This is inspired by the idea of minimum connection cost in evolutionary biology, but we are the first the combine this idea with training neural networks with gradient descent for interpretability.". Regarding quantitative metrics: "Quantitatively, we use Newman's method to compute modularity of network graphs; BIMT achieves the highest modularity for all our test problems."
Q2: Introduction is too general.
A: Agreed! We have now expanded on motivation, literature and more details of the method in the introduction.
Q3: The lack of reference to other methods.
A: Thanks! Now we have added these references in the experiment section.
Q4: Link to the literature of neural codes, neural information transmission/processing, neural spikes.
A: Thanks for pointing us to these references! We have included the discussion in the related works part under "Neuroscience-inspired learning".
Q5 & 6: Discussion and is needed. What are practical implications?
A: Thanks for the suggestion! We have now highlighted broader impacts and limitations in the conclusion & discussion section.
Q7: The figures are of poor quality.
A: We feel sorry for that and have updated figures to higher resolution. We also want to kindly note that different PDF readers may result in different rendering quality and speed. We would recommend Adobe Acrobat PDF reader for the best experience.
Reviewer 3 Report
Comments and Suggestions for Authors
Summary
the authors design a loss function that constrains nodes in a neural network to group together and form modules. They penalize long-distance strongly-weighted connections. In addition, they assist training with intermediate node-swapping steps to help 'untangle' the network. The resulting networks are highly modular, and sparse, allowing for interpretation 'at a glance'. The authors demonstrate their loss function, and ability to interpret the resulting networks, by solving a wide range of test problems.
Feedback
I enjoyed reading this paper. The authors take a concept widely used in neuro-evolution algorithms (penalizing connection lengths) and apply it to the gradient descent training method more familiar in the machine learning sphere. The resulting networks resemble those evolved with connection-length penalties.
The authors' mainly focus on the interpretability of the resulting networks, and they demonstrate that these networks are easily interpreted by correlating the structure of the networks to the properties of the tasks they solve. This lends support to the notion that one could inspect the network and infer properties of the task. Although the modular networks perform worse, analyzing the network structures provides a surprising number of insights, insights that would no doubt be valuable to scientists and engineers looking to solve a specific task.
The authors also identify my main concerns with the work. The scale of the tasks examined in this work are small. It is not clear that the training procedure, and in particular the node embedding and swapping procedure, will scale to larger networks. Additionally, the main focus of the work, interpreting networks, already becomes more difficult with the largest networks presented in this work (MNSIT), and is likely only going to become harder with more complex networks. I believe more work is needed to fully explore the feasibility of this technique to make deep neural networks interpretable. Figure A5, and the related discussion of small perturbations to the initial conditions also raises concerns about the feasibility of this technique to scale; as networks grow larger I would expect the range of outcomes to increase as well, diminishing any hope of finding an average or typical modular structure in large networks across training replicates.
Still, despite these concerns, this work is a good first step towards exploring connection costs in deep neural networks, and how the resulting structures improve interpretability.
Author Response
We would like to thank the reviewer for their time and constructive feedback. We are very happy that the reviewer likes the paper! We understand concerns raised by the reviewer about scalability and replicability. We now include brief plans for future works on how to address these two issues in conclusion.
"To improve scalability, maybe we need to explicitly introduce some notion of hierarchy (beyond modularity), because hierarchy can potentially decompose complicated modules into simpler submodules, hence making scaling easier. To improve replicability, some theoretical research are first needed to characterize equivalent classes of neural networks, so that we can better understand whether the changes caused by perturbation are superficial (in the same equivalent class) or fundamental (in different equivalent classes)."
Round 2
Reviewer 1 Report
Comments and Suggestions for Authors
The authors have revised the manuscript successfully for the parts I showed some concerns. Therefore, I recommend the paper to be published in the current form.